# Latent Gaussian Activity Propagation: Using Smoothness and Structure to Separate and Localize Sounds in Large Noisy Environments

**Daniel D. Johnson, Daniel Gorelik, Ross Mawhorter, Kyle Suver, Weiqing Gu**
Department of Mathematics
Harvey Mudd College
Claremont, CA 91711
{ddjohnson, dgorelik, rmawhorter, ksuver, gu}@hmc.edu

**Steven Xing, Cody Gabriel, Peter Sankhagowit**
Intel Corporation
Hillsboro, OR 97124
{steven.xing, cody.gabriel, peter.sankhagowit}@intel.com

## Abstract

We present an approach for simultaneously separating and localizing multiple sound sources using recorded microphone data. Inspired by topic models, our approach is based on a probabilistic model of inter-microphone phase differences, and poses separation and localization as a Bayesian inference problem. We assume sound activity is locally smooth across time, frequency, and location, and use the known position of the microphones to obtain a consistent separation. We compare the performance of our method against existing algorithms on simulated anechoic voice data and find that it obtains high performance across a variety of input conditions.

## 1 Introduction

Traditional playback of real-world events is usually constrained to the viewpoints of the original cameras and microphones, which limits the immersion of the experience. In contrast, if those events are reconstructed in virtual space, they can be played back from perspectives without a corresponding source recording and explored interactively. Such technology would enable users to experience events in virtual reality with full freedom of motion.

Realistically reproducing the audio of a physical event in virtual space requires that sounds be both faithfully separated from each other and accurately localized. Furthermore, in order to capture events that happen over a large region in space, it is necessary to place microphones far away from each other, which introduces non-negligible delays into the audio signals. As such, we focus on the task of performing blind separation and localization in the presence of noise, where relatively few microphones are placed far away from each other to cover a large area. This situation poses difficulties for some existing separation and localization algorithms, which often assume large numbers of microphones [Dorfan et al., 2015], an array of closely spaced microphones [Mandel et al., 2009, Lewis, 2012], or known sound characteristics [Oldfield et al., 2015, Wang and Chen, 2017].

Specifically, we examine the situation of sound sources in a large, known, anechoic two-dimensional space, with pairs of two omni-directional microphones placed at arbitrary points and orientations. The presence of noise, as well as the potentially large distances between microphone pairs, means that sources may not perfectly correspond across recordings, making separation an ill-posed problem.

We propose a method for simultaneously separating and localizing sounds by performing Bayesian maximum a posteriori inference in an approximate probabilistic model of sound propagation. We model the phase differences between the microphones in each pair as arising from a latent source activity array. A Gaussian process prior is used to ensure that it is locally smooth across time, frequency, and spatial extent, and microphone locations are handled using a set of propagation transformations. This approach is capable of separating sounds in a noisy environment, regardless of the number of sources present, and localizing each source to a subset of a grid of possible locations.

Our algorithm makes the window-disjoint orthogonality assumption (W-DO), i.e., that the time-frequency representations of the sound sources do not overlap [Jourjine et al., 2000]. This allows us to model each element of the time-frequency spectrum as being generated by exactly one source at a specific location. In particular, we model the phase difference as arising from a mixture of von Mises distributions, with each distribution corresponding to the phase differences for an individual location.

We evaluate our method on synthetic data composed of multiple speakers located in a large space with added white noise, and compare the results against those achieved by MESSL [Mandel et al., 2009] and DALAS [Dorfan et al., 2015] for variable numbers of sources and microphone configurations.

## 1.1 Background Information

Our approach – along with many other source separation algorithms – uses the Short-Time Fourier Transform (STFT) representation to analyze the different sounds present in a signal. The STFT is the result of applying the Fourier Transform to short overlapping time windows, and gives a representation of what frequencies are present in a signal in those windows.

Based on the speed of sound, there is a delay in the sound propagation as each source propagates to each microphone. This produces a phase difference between the sounds recorded by the microphones in each pair. Since the pair of microphones is sufficiently close, the phase delay for a given source frequency at a given time is the multiplicative factor $e^{i\omega\delta}$, where $\omega$ is the sound frequency, and $\delta$ is the arrival delay between the two microphones [Rickard, 2007].

The mapping from delay to phase difference is invertible when the distance between the pair of microphones is bounded by $\pi c/\omega_m$, where $\omega_m$ is the maximum source frequency and $c$ is the speed of sound [Rickard, 2007]. This gives an estimate of the true location of the active source. For larger separations and higher frequencies, multiple delays correspond to the same phase difference, making the correct location ambiguous, and requiring the use of additional assumptions to correctly separate sounds on the basis of phase differences.

## 1.2 Prior Work

Blind source separation is a well-studied problem. However, many existing techniques (such as Non-negative Matrix Factorization [Virtanen, 2007], Beamforming [Lewis, 2012], Independent Component Analysis [Hyvärinen et al., 2004, Bell and Sejnowski, 1995, Yeredor, 2001], deep-learning based approaches [Wang and Chen, 2017], and a heuristic sound-specific approach by Oldfield et al. [2015]) make strong assumptions about the spectral structure of the sources, geometry of the microphones, and level of noise, which make them unsuitable for separating our mixtures of interest.

The Degenerate Unmixing Estimation Technique (DUET) focuses on separating degenerate mixtures of sparse sounds that occur across space from two recordings [Rickard, 2007]. It assumes that the sounds do not overlap significantly in the STFT time-frequency representation, so that different sources are dominant at different frequencies at each moment in time. This assumption is known as window-disjoint orthogonality (W-DO). DUET uses the phase differences in the STFT across microphones to estimate the time difference of arrival between the two microphones for that source, which can be used to approximate the direction from which the sound arrived. It then clusters these directions to construct a series of masks that can isolate a large number of spatially separated sounds given only two side-by-side recordings. However, this approach is limited in that it assumes a noiseless environment, and can only accurately estimate time differences for low frequencies due to the aliasing effect of high-frequency waveforms.

The Model-based EM Source Separation and Localization (MESSL) algorithm, proposed by Mandel et al. [2009], builds upon DUET by using a probabilistic model that predicts phase differences given true source delays, and separates sounds by performing maximum-likelihood estimation of

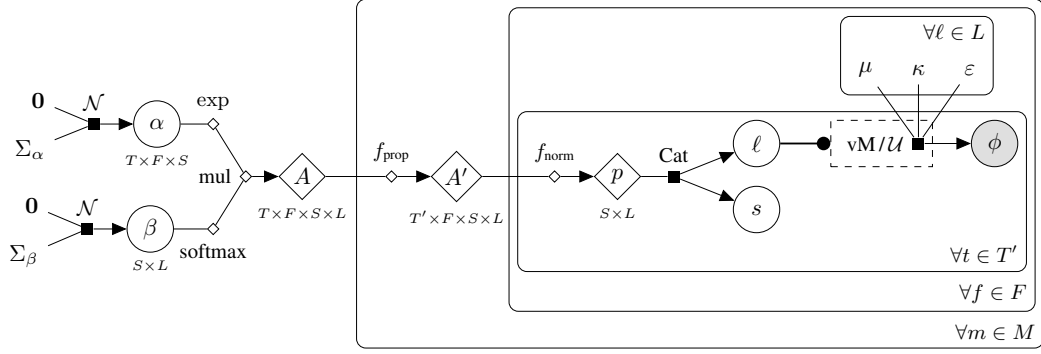

Figure 1: Directed factor graph diagram [Dietz, 2010] of the LGAP model. Circles represent random variables, diamonds represent deterministic functions of parent variables, and unmarked symbols denote hyperparameters. Along edges, small black squares represent directed factors (i.e., conditional probability distributions of children given parents), small white diamonds represent function application, and the small black circle represents a choice "gate" (see Dietz [2010]). Plate notation indicates sets of independent variables, and array-valued variables are labeled below with their index sets. $\mathcal{N}$ denotes a multivariate normal distribution, Cat denotes a categorical (multinomial) distribution, and vM/$\mathcal{U}$ represents the von Mises-Uniform mixture described in Section 2.3.

the delay parameters. This approach avoids the aliasing problem, as the mapping from true delay to phase difference is one-to-one. From this premise they identify the parts of the spectrogram of the signal which best fit models being constructed of the mixture using an Expectation-Maximization (EM) method. Although the original algorithm focuses on a single microphone pair and assumes each time-frequency component is independent, MESSL has been extended to incorporate local smoothness using a Markov random field [Mandel and Roman, 2015] and to work with more than one microphone pair [Bagchi et al., 2015]. However, these extensions focus on the use of MESSL for separating speech mixtures in small environments, and were not designed for use in large noisy environments with distant microphones.

The Distributed Algorithm for Localization and Separation (DALAS) extends MESSL to work with spatially-separated microphones in a known configuration [Dorfan et al., 2015]. The first step in this approach is to run an incremental distributed expectation-maximization (IDEM) algorithm to find the maximum likelihood estimate of the location distribution of the sources, using the known configuration of microphones to model the possible phase differences associated with each spatial position. Next, it associates each peak of the localization distribution with a source, and matches each source to its closest microphone pair. Finally, a spectral mask is created using thresholding and the node values are then filtered to give the separated sources. This technique has the advantage of working with spatially separated microphones over a large area. However, it was not designed for noisy environments, and assumes that every time-frequency component is independent, ignoring the temporal and frequential structure of each sound source.

## 2 Probabilistic Model

We cast source separation and localization as a Bayesian inference problem. We model each time-frequency bin as being assigned a latent dominating source and location that determines the distribution of each observed phase difference. These assignments are in turn drawn proportional to a smooth latent activity array $A$, which causes the assignments for nearby observations to be correlated. To account for arrival time differences between microphones, this latent activity is corrected for each microphone using a propagation function before being used to determine the dominating locations. Using this model, which we call Latent Gaussian Activity Propagation (LGAP), sounds can then be separated by performing Bayesian inference on the latent source and location assignments.

Let $T$ be a set of time bins, $F$ a set of frequency bins, $L$ a set of candidate locations, and $M$ a set of microphone pairs. A directed factor graph diagram [Dietz, 2010] of the LGAP model is shown in Figure 1. We assume the sound was generated by a small set of sources $S$, and let

$A \in \mathbb{R}^{|T| \times |F| \times |S| \times |L|}$ be an array-valued random variable such that $A_{t,f,s,\ell}$ is proportional to the likelihood of hearing a sound from source $s$ at location $\ell$, time $t$, and frequency $f$. We also assume that the location of each source does not change with time or frequency, and thus express $A$ as a product of time-frequency activity and spatial extent terms, denoted $\alpha \in \mathbb{R}^{|T| \times |F| \times |S|}$ and $\beta \in \mathbb{R}^{|S| \times |L|}$.

This activity array is propagated through time for each microphone according to location-dependent time delays, resulting in a propagated location-activity array for each microphone. Next, each time-frequency bin of the observed audio is assigned a single dominating source and location, proportional to the propagated activity of each. Finally, the phase difference observations are sampled from a location-specific phase distribution.

## 2.1 Sound Activity

We encourage the latent activity array to be smooth by imposing Gaussian process priors over the random variables $\alpha$ and $\beta$. In particular, we interpret each element of $\alpha$ and $\beta$ as evaluations of a function over the index sets for those variables, i.e., we interpret $\alpha \in \mathbb{R}^{|T| \times |F| \times |S|}$ as a random function $\alpha : T \times F \times S \rightarrow \mathbb{R}$ such that $\alpha(t, f, s)$ yields the activity of source $s$ at time $t$ and frequency $f$, and specify the covariance matrix $\Sigma_\alpha$ for $\alpha$ by evaluating a chosen Gaussian process kernel over the set $T \times F \times S$ of possible times, frequencies, and sources, yielding multivariate normal distribution $\alpha \sim \mathcal{N}(\mathbf{0}, \Sigma_\alpha)$. By choosing these kernels to favor smooth functions, we encode our prior belief that the latent activity arrays are smooth over specific time scales, frequency ranges, or spatial areas.

In practice, $|T|$, $|F|$, and $|L|$ may be very large, leading to memory and computation issues. To efficiently enforce our smoothness priors over high dimensional spaces, we choose Gaussian process kernels that factorize into combinations of axis-specific (positive definite) kernel functions, i.e.,

$$k_\alpha\Big((t_1, f_1, s_1), (t_2, f_2, s_2)\Big) = k_T(t_1, t_2)\, k_F(f_1, f_2)\, k_{S_1}(s_1, s_2) + k_{S_2}(s_1, s_2), \qquad (1)$$

$$k_\beta\Big((s_1, \ell_1), (s_2, \ell_2)\Big) = k_L(\ell_1, \ell_2) k_{S_1}(s_1, s_2), \qquad (2)$$

Computations involving the resulting multivariate normal distributions require factorizing each covariance matrix as $\Sigma = SS^T$ (where $S$ is called a *scale* matrix) and multiplying vectors by these matrices. Fortunately, the axis-aligned structure of our kernels makes these multiplications efficient.

**Proposition 1.** *Suppose* $\Sigma_\alpha : \mathbb{R}^{(|T| \times |F| \times |S|) \times (|T| \times |F| \times |S|)}$ *and* $\Sigma_\beta : \mathbb{R}^{(|S| \times |L|) \times (|S| \times |L|)}$ *are covariance matrices arising from the evaluation of kernels (1) and (2) on a grid of points. There exist factorizations* $\Sigma_\alpha = S_\alpha S_\alpha^T$, $\Sigma_\beta = S_\beta S_\beta^T$ *such that multiplying vectors by* $S_\alpha$ *and* $S_\beta$ *(and their inverses) can be performed in* $O\big(|T|\,|F|\,|S|\,(|T| + |F| + |S|)\big)$ *and* $O\big(|S|\,|L|\,(|S| + |L|)\big)$ *arithmetic operations, respectively.*

Specific kernels for $k_T$, $k_F$, and $k_L$ can be chosen based on prior knowledge about the application domain. For our experiments, we use rational quadratic kernels with length-scales of 0.1 seconds and 1000 Hz for $k_T$ and $k_F$, respectively, with the $\alpha$ parameter set to 0.1, and use a mixture of two equally-likely radial basis function kernels for $k_L$, with length scales of 3/32 and 5/8 times the size of our test region.

We assume that individual sources are independent, i.e., we define $k_{S_1}(s_1, s_2) = \delta_{s_1 s_2}$ to avoid enforcing any structure across sources. Additionally, we set $k_{S_2}(s_1, s_2) = C\delta_{s_1 s_2}$, where $C$ is chosen to account for the magnitude of differences in average activity between sources. In our experiments, we set $C = 3$ and similarly scale up the rational quadratic and radial basis function kernels to have maximum value 3, as we found that this produced a plausible prior distribution of output assignments after normalization (described in Section 2.3).

Since we interpret the activity matrix $A$ as being proportional to the contribution of each source and location, we need to ensure that its elements are nonnegative. We thus compute

$$A_{t,f,s,\ell} = \exp(\alpha_{t,f,s}) \frac{\exp(\beta_{s,\ell})}{\sum_{\ell'} \exp(\beta_{s,\ell'})}.$$

Note that the $\beta$ contribution is normalized across locations for each source, which prevents sources that are spread across multiple locations (or have uncertain location estimates during inference) from being proportionally "louder" and dominating the final source assignments.

## 2.2 Propagation

In a large environment, it is likely that sounds arrive at each microphone pair with a delay greater than the resolution of our STFT time bins. The activity matrix $A$ must thus be corrected for each microphone pair location, by shifting activity from far locations forward in time so that they are heard after the appropriate delay.

For a fixed location $\ell$ and microphone pair $m$, let $\tau_{m,\ell}$ denote the propagation time delay between a sound wave leaving $\ell$ and arriving at $m$, and let $\overline{\tau}_\ell$ denote a reference delay for each location. We compute $A'_m = f^m_{\text{prop}}(A)$ as

$$A'_{m,t_2,f,s,\ell} = \sum_{t_1 \in T} w(t_2 - t_1 - \tau_{m,\ell} + \overline{\tau}_\ell) A_{t_1,f,s,\ell},$$

where $w : \mathbb{R} \to \mathbb{R}$ is a weighting function that peaks at zero. This shifts the values of $A$ along the time axis by $\tau_{m,\ell} - \overline{\tau}_\ell$ bins, the delay (in STFT frames) of microphone $m$ relative to the reference delay for a sound at location $\ell$, and also blurs it slightly across bins to account for uncertainty in location and timing or misalignment between discrete timesteps. This operation can be efficiently implemented as a convolution operation with a precomputed convolution filter. In practice, we let $\overline{\tau}_\ell = \tau_{m^\star,\ell}$ for some chosen reference microphone pair $m^\star$, and choose $w$ based on finite differences of a logistic sigmoid function

$$w(t) = \begin{cases} \frac{1}{1+\exp(-(t+0.5)/\sigma)} - \frac{1}{1+\exp(-(t-0.5)/\sigma)} & -3\sigma \leq t \leq 3\sigma, \\ 0 & \text{otherwise,} \end{cases}$$

where $\sigma$ is chosen based on how close points in $L$ are each other (i.e., the delay uncertainty due to our discretization of the region).

Due to the different offsets for each microphone, sounds that arrive at the microphones over a particular time interval may be explained by activity that occurs outside of that interval. The propagation transformation thus transforms our original set of timesteps $T$ for our latent activity to a slightly smaller set of timesteps $T'$ corresponding to the observed data.

## 2.3 Assignments and Observations

After propagation, we normalize the activity array $A'$ across sources and locations, obtaining source-location probabilities

$$p_{m,t,f,s,\ell} = \frac{A'_{m,t,f,s,\ell}}{\sum_{s' \in S, \ell' \in L} A'_{m,t,f,s',\ell'}}.$$

We then sample latent source and location assignments for each microphone pair $m$, time $t$, and frequency $f$, representing the source and location that dominated the recording for that microphone pair at that moment. In particular, we sample $s_{m,t,f}, \ell_{m,t,f} \sim \text{Cat}(p_{m,t,f})$, so that

$$P(s_{m,t,f} = s \wedge \ell_{m,t,f} = \ell) = p_{m,t,f,s,\ell}.$$

Given the location $\ell_{m,t,f}$ that dominates a given time-frequency bin, we can predict the phase difference $\phi_{m,t,f}$ received by a given microphone pair $m$. Let $\Delta_{m,\ell}$ denote the difference between the distance of the first microphone in the pair to $\ell$ and the distance of the second microphone to $\ell$, $\delta_{m,\ell}$ denote our uncertainty in the value of $\Delta_{m,\ell}$ due to our discrete set of locations, and $c$ denote the speed of sound. We model $\phi_{m,t,f}$ as being drawn from a von Mises distribution $\text{vM}(\mu_{m,f,\ell}, \kappa_{m,f,\ell})$ with probability $1 - \epsilon_{m,f,\ell}$, and from $\mathcal{U}(0, 2\pi)$ with probability $\epsilon_{m,f,\ell}$, where

$$\mu_{m,f,\ell} = 2\pi f \frac{\Delta_{m,\ell}}{c} \mod 2\pi, \qquad \kappa_{m,f,\ell} = \left(2\pi f \frac{\delta_{m,\ell}}{c}\right)^{-2}, \qquad \epsilon_{m,f,\ell} = 0.001.$$

For efficiency, we group the individual raw STFT bins into small regions, and assume all observations within each region were generated by the same von Mises/Uniform mixture. This allows our $T$ and $F$ sets to be smaller than the number of true STFT components computed while still using all available phase information.

## 2.4 Inference

Given a set of known phase observations $\phi$, we can separate and localize the sounds by performing Bayesian inference on the assignments $s_{m,t,f}$ and $\ell_{m,t,f}$ for each of our data points. We focus on approximating the maximum a posteriori (MAP) estimate of the parameters, i.e., the most likely separation given our observations.

We approximate the MAP solution by marginalizing out $s_{m,t,f}$ and $\ell_{m,t,f}$ and performing gradient ascent on the log posterior

$$\log P(\alpha, \beta | \phi) = \log P(\alpha) + \log P(\beta) + \log P(\phi | \alpha, \beta) - \log P(\phi).$$

Note that, due to our smoothness assumption, the covariance matrices $\Sigma_\alpha$ and $\Sigma_\beta$ are poorly conditioned, with small eigenvalues along directions of rapid oscillation. These values can hamper convergence of gradient descent, as the log-posterior term contains products with the inverse covariance matrices, leading to steep valleys in the loss landscape. We thus perform gradient descent in the eigenspace of the covariance matrices, and use a preconditioning matrix to scale the small eigenvalues so that the minimum eigenvalue of the system is 1. We also employ gradient clipping to prevent the algorithm from diverging. For computational efficiency, we only consider a random subset of frequencies at each iteration of gradient descent, since this gives an unbiased estimate of the true gradient.

After obtaining estimates of $\alpha$ and $\beta$, we can then estimate the posterior probabilities $P(s_{m,t,f}, \ell_{m,t,f} | \alpha, \beta, \phi)$ by computing

$$P(s_{m,t,f}, \ell_{m,t,f} | \alpha, \beta, \phi) \propto P(s_{m,t,f}, \ell_{m,t,f} | \alpha, \beta) P(\phi | \ell_{m,t,f}).$$

These probabilities can be used to assign portions of the input to specific sources and locations.

## 2.5 Garbage Source

Microphone-specific noise, reverberations, and interactions between sources can produce phase observations that are not consistent with any true source location. To enable the model to handle these non-localized components, we follow Mandel et al. [2009] and designate one of the sources $s_g \in S$ as a "garbage source". This source is constrained so that, instead of being assigned to a specific 2D coordinate, it is instead assigned to a "garbage location" $\ell_g \in L$ with no associated time delay ($\Delta_{m,\ell_g} = 0$), and whose corresponding phase distribution is uniform over $[0, 2\pi)$ (so $\epsilon_{m,f,\ell_g} = 1$). When using a garbage source, we let $\beta$ range only over $S \setminus \{s_g\}$ and $L \setminus \{\ell_g\}$, and compute

$$A_{t,f,s,\ell} = \begin{cases} \exp(\alpha_{t,f,s}) \frac{\exp(\beta_{s,\ell})}{\sum_{\ell' \neq \ell_g} \exp(\beta_{s,\ell'})} & \ell \neq \ell_g, s \neq s_g, \\ \exp(\alpha_{t,f,s}) & \ell = \ell_g, s = s_g, \\ 0 & \text{otherwise,} \end{cases}$$

i.e., we constrain $s_g$ to assign all of its activity to $\ell_g$ and all other sources to assign none to $\ell_g$.

## 2.6 Initialization

The LGAP model is fairly sensitive to the initialization of the parameters $\alpha$ and $\beta$. Following Mandel et al. [2009], we start by computing the PHAT-histogram [Aarabi, 2002] of the input for each microphone, and then identify peaks in the histogram to estimate potential source time delays. In our case, we choose the microphone pair with the most cleanly-identified peaks as $m^\star$, and use the peaks for that pair to initialize our source activity array $\alpha$. In particular, we compute the likelihood of each phase observation under a set of von Mises distributions centered at each peak (along with a uniform distribution for the garbage source), then smooth these likelihoods over time and frequency using a Gaussian blur, and initialize $\alpha$ from the logarithm of these smoothed likelihoods. This ensures that our initial propagated activity array $A'$ assigns each of the peaks of the reference microphone to a distinct source. $\beta$ is initialized uninformatively to an array of zeros.

To optimize our parameters from this initialization, we start by holding $\alpha$ fixed and running gradient ascent on our posterior probabilities with respect to $\beta$. This gives us an initial estimate of the location of each of the sources identified from our initialization. Next, we hold $\beta$ constant, reset $\alpha$ to zero, and run gradient ascent on $\alpha$ to re-estimate our time-frequency masks to be consistent with the identified locations without being biased by the approximate PHAT-histogram masks. Finally, we perform gradient ascent for a longer duration on both $\alpha$ and $\beta$ to fine-tune our full activity matrix.

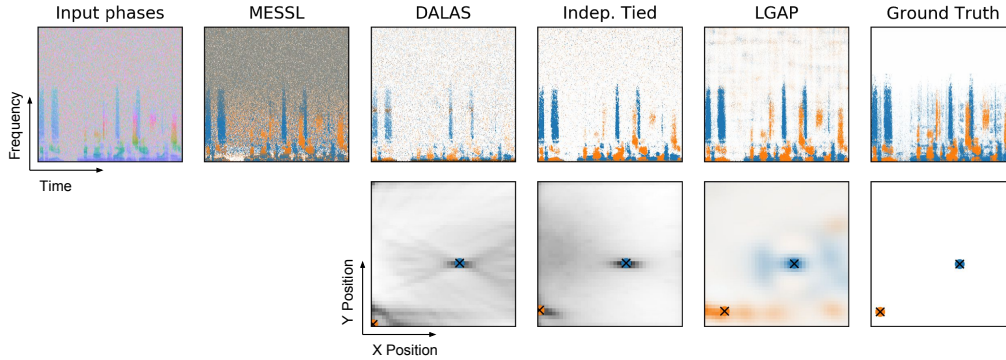

Figure 2: Generated masks for one of our test examples, consisting of two sources recorded by five microphone pairs over a 32 m by 32 m square. Top row: separated spectrograms of the recordings from the center microphone pair, over four seconds (horizontal axis) across frequencies up to 22050 Hz (vertical axis). Bottom row: location estimates, where shading represents confidence and crosses represent predicted point locations. For LGAP, location estimates are colored by source, since LGAP estimates a separate location distribution for each.

## 3  Experiments

We evaluate our algorithm on a set of synthetic anechoic speech mixtures using audio from the Mozilla Common Voice Dataset[1]. For each test, we place sound sources randomly within a square region, and compute the waveforms received by each of the microphones, which are arranged parallel to one side of this region with a pair spacing of 5 cm. To prevent unrealistic phase observations due to inaudible contributions of sources at high frequencies, and to demonstrate robustness against noise, we add a small amount of white noise to each of our microphones.

For our method, we use rational quadratic kernels for our time and frequency kernels and a mixture of radial basis functions for our location kernel, as described in Section 2.1. We enable the garbage source to handle the additional microphone noise, and group STFT bins into rectangular regions consisting of 8 timesteps and 2 frequencies for computational efficiency.

We compare our method with a number of baselines:

- MESSL [Mandel et al., 2009, Mandel and Roman, 2015]: Although MESSL has been extended to work with many-channel recordings [Bagchi et al., 2015], it assumes all microphones are adjacent and does not give location estimates. We thus run MESSL individually on each of our microphone pairs, using MRF smoothing and a garbage source. We do not incorporate level differences, as these are not used by the other methods and are intended for use with a dummy head placed between the pairs, which we do not simulate here.
- DALAS [Dorfan et al., 2015]: For consistency with our method, we modify DALAS to use an identical von Mises distribution of phase offsets for each location, and select a fixed number of sources to separate instead of choosing it dynamically.
- Independent tied-microphone model: A simple baseline, based on the DUET algorithm, that assumes that the corresponding time-frequency bin for each microphone pair was generated by a single location $\ell_{m,t,f} = \ell_{t,f}$ (i.e., the same source dominates across all microphones, ignoring time delays and noise), and that these locations were independently chosen and uniformly distributed (ignoring smoothness over time and frequency). We calculate the posterior distribution over these locations explicitly as

$$P(\ell_{t,f}|\phi_{1,t,f},\ldots,\phi_{M,t,f}) \propto P(\ell_{t,f}) \prod_{m=1}^{M} P(\phi_{m,t,f}|\ell_{t,f}) \propto \prod_{m=1}^{M} P(\phi_{m,t,f}|\ell_{t,f})$$

where $P(\phi_{m,t,f}|\ell_{t,f})$ is the same von Mises-Uniform mixture described in Section 2.3. We then choose a fixed number of locations with the highest likelihood averaged across all bins and use those as the sources to extract.

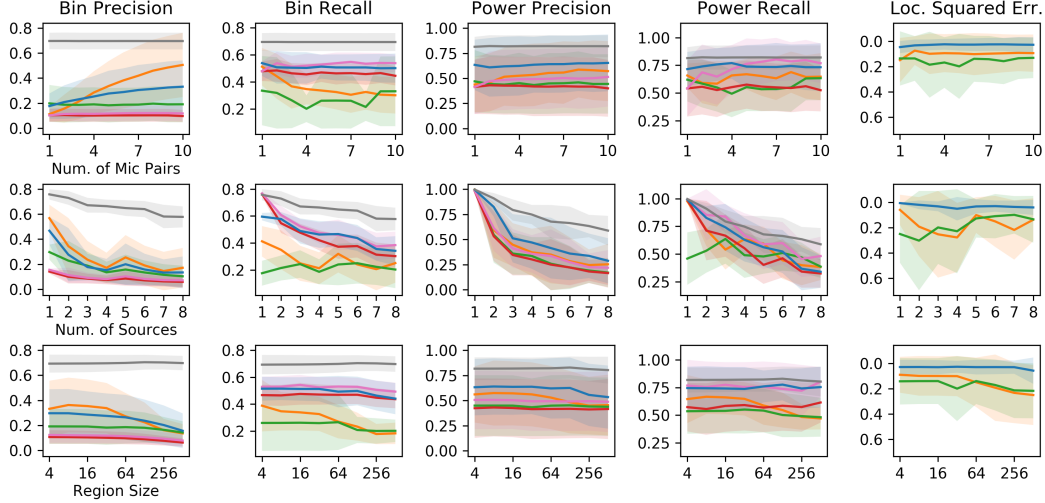

Figure 3: Experimental results for each method: — LGAP, — Indep. Tied, — DALAS, — MESSL (All mics), — MESSL (Best mic), — Ideal Mask. Lines represent mean performance and shaded regions represent one standard deviation, computed across all sources separately for each of five trials. Location squared error is reported relative to the side length of the square region. Region size is in meters. MESSL performance is reported averaged across all microphones as well as for the single microphone pair with the best separation. For reference, Ideal Mask gives the performance of the ideal ratio mask [Srinivasan et al., 2006].

For all methods, we compute the STFT using a Hann window with a frame size of 512 samples and a step of 128 samples (for a 44100 Hz signal). Additionally, for all methods except MESSL, we specify the set $L$ of possible locations as a 40 by 40 grid covering this square region, along with a garbage location. (The DALAS and independent-tied methods do not process this garbage location separately, and instead simply treat it as another possible location.) For the DALAS and independent-tied methods, after selecting our set of sources, we re-normalize the masks across those sources so that mask values for each time-frequency bin sum to one, effectively conditioning on that bin being generated at one of the (fixed number of) most likely source locations. We then divide by the maximum mask value for each source to amplify sources detected with low confidence. See Figure 2 for an example of the resulting masks and location estimates corresponding to each method.

We evaluate the performance of each algorithm on a suite of tests, varying three parameters: the number of microphone pairs (from 1 to 10), the number of sources (from 1 to 8), and the size of the square region (from 4 m to 512 m along each side). We conduct five experiments for each set of parameters, varying each parameter while holding the others at default values (5 microphone pairs, 3 sources, 32 m).

To evaluate the localization performance of each method, we compute the squared error of the estimate, normalized by the size of the square region. To evaluate separation performance, we compare the masks produced by each algorithm to the ideal ratio mask [Srinivasan et al., 2006], which assigns bins to sources proportionally to the ratio of the power spectral density of each source. Letting $R$ denote the ratio mask, $M$ denote a separation method's mask, and $P$ denote the total power spectral density of the recording, we compute *bin precision*, *bin recall*, *power precision*, and *power recall* for a source $s$ as

$$\text{BP}_s = \frac{\sum_{m,t,f} M_{m,t,f,s} R_{m,t,f,s}}{\sum_{m,t,f} M_{m,t,f,s}}, \qquad \text{BR}_s = \frac{\sum_{m,t,f} M_{m,t,f,s} R_{m,t,f,s}}{\sum_{m,t,f} R_{m,t,f,s}},$$

$$\text{PP}_s = \frac{\sum_{m,t,f} M_{m,t,f,s} R_{m,t,f,s} P_{m,t,f,s}}{\sum_{m,t,f} M_{m,t,f,s} P_{m,t,f,s}}, \qquad \text{PR}_s = \frac{\sum_{m,t,f} M_{m,t,f,s} R_{m,t,f,s} P_{m,t,f,s}}{\sum_{m,t,f} R_{m,t,f,s} P_{m,t,f,s}},$$

Bin precision quantifies how much of the proposed mask corresponds to the source, and bin recall quantifies how much of the source is recovered by the mask. Power precision and recall are weighted by the audio power, and thus quantify how much of the energy passed through the mask corresponds

to the source and how much of the source energy is recovered by the mask. Note that power precision is a (nonlinear) transformation of the signal-to-interference ratio. Since the true sources have no definite ordering, we assign each true source to the proposed source that attains the highest bin precision. In addition to computing these metrics for each of the methods, we also compute them for the ratio mask itself. Since MESSL was computed for each microphone pair separately, we report both the mean performance across microphone pairs and the performance on the microphone pair for which MESSL performs the best.

Figure 3 shows the results of our experiments. We see that LGAP is competitive with the other methods across all metrics. Note that the independent tied-microphone method has high precision, but low recall, as its strong assumptions cause it to erroneously classify large fractions of the input as noise, whereas MESSL obtains high recall but low precision, indicating that it recovers most of each source but also admits some interference. LGAP, on the other hand, is able to maintain both high precision and high recall. In addition, and of particular importance for virtual reality applications, the LGAP model obtains reliably accurate location estimates, and consistently outperforms the other localization methods across all input conditions.

Interestingly, the independent tied-microphone baseline performs quite well, especially in small regions and with few sources. This suggests that, when propagation delays are small and there are only a few events, independent consideration of the phase shifts at each time-frequency bin is sufficient to obtain good source masks and location estimates. However, as region size grows and more sources are added, the time delays cause different sources to be active at the same time, reducing the effectiveness of this baseline method.

LGAP maintains high performance even with a small number of microphones. In addition, as the number of microphones increases, LGAP is able to combine information across microphones and improve its precision markedly. As the number of sources are increased, all methods show decreased performance, but LGAP maintains relatively high precision across large numbers of sources. And as the region size grows, the LGAP method is able to account for the increased propagation delays and maintain high performance.

## 4  Conclusion

We have described a Bayesian method for sound separation and localization that incorporates known microphone positions and smoothness assumptions to improve separation quality. This method is robust to a variety of input conditions, and can combine information from distant microphones even in the presence of significant time delays. Apart from the smoothness assumption, the method does not depend on source statistics. It thus has the potential to be used for a variety of applications, and is particularly suited to capturing audio events that are distributed across large real-world environments.

The experiments and analysis presented here use a simple microphone configuration and synthetic dataset, and focus on how much of the sound is preserved and correctly separated by each method. It would be interesting to compare the methods using other metrics, such as BSS_EVAL, which decomposes the error into interference and artifact components [Vincent et al., 2006], or PEASS, which estimates the subjective quality of the separation using the decomposed error [Emiya et al., 2011]. Also of relevance would be to evaluate the methods using different microphone geometries, types of audio, and distribution of noise. For instance, one could conduct an experiment using diffuse noise that is correlated at low frequencies to mimic more realistic noise fields [Habets and Gannot, 2007], or set up physical microphones to measure performance in a non-synthetic task. These experiments are left to future work.

There is considerable room to extend the LGAP model presented here by imposing different structural restrictions on the latent activity arrays. For instance, it would be possible to explicitly model sources with different spatial, temporal and frequential characteristics by using distinct covariance matrices for each, or modify the activity matrix to handle sources that move over time. Additionally, it would be interesting to explore methods for making inference more efficient, such as by exploring simpler local smoothness priors (instead of a global Gaussian process) or discretizing the set of possible phase observations. The masks generated by LGAP could be post-processed to further improve separation, for instance by using beamforming to combine audio across all microphones. Finally, LGAP could easily be extended to work with more complex microphone geometries instead of pairs, which could improve both separation and localization performance.

## Footnotes

[1] https://voice.mozilla.org/

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
