[Supplementary Material · LGAP_appendix.pdf]



Figure 4: Performance of each method on a five-source input over a 32 m by 32 m region, with time-frequency masks on the left and location estimates on the right (where shading represents confidence and colored crosses represent predicted point locations). Note that the LGAP method is able to localize all five sources to their approximate locations, although the separation of the red and purple sources are mixed for low frequencies. Colors are assigned so that each extracted source is colored identically to the ground-truth source it is paired with (if any).

Figure 5: Performance of each method for a three-source mixture over a 256 m by 256 m region. In this space, significant time delays cause the sources to have different temporal alignments relative to each other at each microphone pair. Note that LGAP is able to compensate for this due to the propagation transformation, whereas the independent tied technique fails to separate events with short duration due to its strong assumption that delays are negligible. Also note that MESSL obtains good separation for some microphone pairs but not others, since it cannot use information across microphones to refine its estimates.

## A   Proofs

To simplify notation, if $A$ is a matrix in $\mathbb{R}^{(n_1 \times n_2 \times \cdots \times n_k) \times (n_1 \times n_2 \times \cdots \times n_k)}$ and $v$ is a vector in $\mathbb{R}^{n_1 \times n_2 \times \cdots \times n_k}$, we will denote the elements of $v$ as $v_{(i_1, i_2, \dots, i_k)}$ and the elements of $A$ as $A_{(i_1, i_2, \dots, i_k), (j_1, j_2, \dots, j_k)}$, where $(i_1, i_2, \dots, i_k)$ can be interpreted as shorthand for the index $(((i_1 n_2 + i_2) n_3 + i_3) n_4 + \cdots) n_k + i_k$ into the flattened vector of elements.

**Lemma 1.** *If*
$$B = I_{n_1} \otimes A \otimes I_{n_3},$$
*where $A \in \mathbb{R}^{n_2 \times n_2}$ and $\otimes$ denotes the Kronecker product, then $B$ can be multiplied by a vector $v \in \mathbb{R}^{n_1 \times n_2 \times n_3}$ in*
$$O\left(n_1 \, n_3 \, n_2^2\right)$$
*arithmetic operations.*

*Proof.* By the definition of the Kronecker product,
$$B_{(i_1, i_2, i_3), (j_1, j_2, j_3)} = (I_{n_1})_{i_1, j_1} \, A_{i_2, j_2} \, (I_{n_3})_{i_3, j_3} = \begin{cases} A_{i_2, j_2} & i_1 = j_1, i_3 = j_3, \\ 0 & \text{otherwise.} \end{cases}$$

We can thus compute
$$(Bv)_{(i_1, i_2, i_3)} = \sum_{j_1, j_2, j_3} B_{(i_1, i_2, i_3), (j_1, j_2, j_3)} \, v_{(j_1, j_2, j_3)} = \sum_{j_2 = 1}^{n_2} B_{(j_1, i_2, j_3), (j_1, j_2, j_3)} \, v_{(j_1, j_2, j_3)}.$$

Each element of $Bv$ can thus be computed using $O(n_2)$ operations, so since there are $n_1 n_2 n_3$ elements of $Bv$, the full product can be computed in $O\left(n_1 \, n_3 \, n_2^2\right)$ operations as desired. $\qquad \square$

**Lemma 2.** *If a matrix $A \in \mathbb{R}^{(n_1 \times n_2 \times \cdots \times n_k) \times (n_1 \times n_2 \times \cdots \times n_k)}$ can be expressed as the Kronecker product*
$$A = A^{(1)} \otimes A^{(2)} \otimes \cdots \otimes A^{(k)},$$
*of smaller matrices $A^{(i)} \in \mathbb{R}^{n_i \times n_i}$, then $A$ can be multiplied by a vector $v \in \mathbb{R}^{n_1 \times n_2 \times \cdots \times n_k}$ in*
$$O\left(n_1 n_2 \dots n_k (n_1 + n_2 + \cdots + n_k)\right)$$
*arithmetic operations.*

*Proof.* By the mixed-product property of Kronecker-factored matrices, we know
$$A = (A^{(1)} \otimes I_{n_2} \otimes \cdots \otimes I_{n_k})(I_{n_1} \otimes A^{(2)} \otimes \cdots \otimes I_{n_k}) \cdots (I_{n_1} \otimes I_{n_2} \otimes \cdots \otimes A^{(k)})$$
$$= (I_1 \otimes A^{(1)} \otimes I_{n_2 n_3 \dots n_k})(I_{n_1} \otimes A^{(2)} \otimes I_{n_3 n_4 \dots n_k}) \cdots (I_{n_1 n_2 \dots n_{k-1}} \otimes A^{(k)} \otimes I_1).$$

By Lemma 1, we can compute the product of a vector with each of the matrices
$$I_{n_1 \dots n_{i-1}} \otimes A^{(i)} \otimes I_{n_{i+1} \dots n_k}$$
in $O(n_1 n_2 \dots n_k n_i)$ arithmetic operations. We can calculate $Av$ by multiplying $v$ by each of these matrices in sequence, for an overall runtime of
$$O\left(n_1 n_2 \dots n_k (n_1 + n_2 + \cdots + n_k)\right)$$
arithmetic operations, as desired. $\qquad \square$

**Lemma 3.** *Let $A \in \mathbb{R}^{n \times n}$ be an invertible matrix such that we can compute $Av$ and $A^{-1}v$ (for $v \in \mathbb{R}^n$) in $O(\alpha n)$ operations, and $B$ be a positive semidefinite matrix with rank $r$. Further suppose that $AA^T + B$ is invertible. Then there exists a matrix $C \in \mathbb{R}^{n \times n}$ with*
$$CC^T = AA^T + B$$
*such that we can compute $Cv$ and $C^{-1}v$ in $O(\alpha n + rn)$ operations.*

*Proof.* Since $B$ is positive semidefinite with rank $r$, we can write $B = UDU^T$ where $U \in \mathbb{R}^{n \times r}$ has orthonormal columns and $D = \text{diag}(\lambda_1, \ldots, \lambda_r)$. Note then that

$$
\begin{aligned}
AA^T + B &= AA^T + UDU^T \\
&= A\big(I_n + A^{-1}UDU^T(A^T)^{-1}\big)A^T \\
&= A\big(I_n + (A^{-1}U)D(A^{-1}U)^T\big)A^T.
\end{aligned}
$$

Let $X = A^{-1}U \in \mathbb{R}^{n \times r}$. Using the singular value decomposition, we can write $X = VEW^T$ where $V \in \mathbb{R}^{n \times r}, W \in \mathbb{R}^{r \times r}$ have orthonormal columns and $E \in \mathbb{R}^{r \times r}$ is diagonal. Then

$$
\begin{aligned}
AA^T + B &= A\big(I_n + XDX^T\big)A^T \\
&= A\big(I_n + VEW^TDWEV^T\big)A^T.
\end{aligned}
$$

Now let $Y = EW^TDWE \in \mathbb{R}^{r \times r}$ and note that $Y$ is symmetric. It can thus be orthogonally diagonalized as $Y = PFP^T$ with $P \in \mathbb{R}^{r \times r}$ having orthogonal columns and $F = \text{diag}(\mu_1, \ldots, \mu_r) \in \mathbb{R}^{r \times r}$. Thus

$$
\begin{aligned}
AA^T + B &= A\big(I_n + VYV^T\big)A^T \\
&= A\big(I_n + VPFP^TV^T\big)A^T \\
&= A\big(I_n + (VP)F(VP)^T\big)A^T \\
&= A\big(I_n + QFQ^T\big)A^T.
\end{aligned}
$$

where $Q = VP \in \mathbb{R}^{n \times r}$ has orthonormal columns. Next let

$$
G = (I_r + F)^{1/2} - I_r = \text{diag}\left(\sqrt{1 + \mu_1} - 1, \ \ldots, \ \sqrt{1 + \mu_r} - 1\right).
$$

We can extend $Q$, $F$, and $G$ into $\widetilde{Q} \in \mathbb{R}^{n \times n}, \widetilde{F} \in \mathbb{R}^{n \times n}, \widetilde{G} \in \mathbb{R}^{n \times n}$ by adding additional orthonormal columns to $Q$ and zeros along the diagonal to $F$ and $G$. Then $QGQ^T = \widetilde{Q}\widetilde{G}\widetilde{Q}^T, QFQ^T = \widetilde{Q}\widetilde{F}\widetilde{Q}^T$, and thus we must have

$$
\begin{aligned}
\big(I_n + QGQ^T\big)\big(I_n + QGQ^T\big)^T &= \big(I_n + QGQ^T\big)^2 \\
&= I_n + 2QGQ^T + QG^2Q^T \\
&= I_n + 2\widetilde{Q}\widetilde{G}\widetilde{Q}^T + \widetilde{Q}\widetilde{G}^2\widetilde{Q}^T \\
&= \widetilde{Q}\left(I_n + 2\widetilde{G} + \widetilde{G}^2\right)\widetilde{Q}^T \\
&= \widetilde{Q}\left(I_n + 2\left((I_n + \widetilde{F})^{1/2} - I\right) + \left((I_n + \widetilde{F})^{1/2} - I_n\right)^2\right)\widetilde{Q}^T \\
&= \widetilde{Q}\left(I_n + 2(I_n + \widetilde{F})^{1/2} - 2I_n + (I_n + \widetilde{F}) - 2(I_n + \widetilde{F})^{1/2} + I_n\right)\widetilde{Q}^T \\
&= \widetilde{Q}\left(I_n + \widetilde{F}\right)\widetilde{Q}^T = I_n + \widetilde{Q}\widetilde{F}\widetilde{Q}^T = I_n + QFQ^T.
\end{aligned}
$$

We can thus let $C = A\big(I_n + QGQ^T\big)$, which ensures that $CC^T = AA^T + B$.

Now consider the product $Cv = A\big(I_n + QGQ^T\big)v$ for $v \in \mathbb{R}^n$. We can compute $q = QGQ^Tv$ in $O(rn)$ operations, add $u = v + w$ in $O(n)$ operations, and compute $Au$ in $O(\alpha n)$ operations, for a total of $O(\alpha n + rn)$ operations.

By the Woodbury matrix identity, we also find that

$$
\begin{aligned}
C^{-1} &= \big(I_n + QGQ^T\big)^{-1}A^{-1} \\
&= \left(I_n - Q\left(G^{-1} + Q^TQ\right)Q^T\right)A^{-1} \\
&= \left(I_n - Q\left(G^{-1} + I_r\right)Q^T\right)A^{-1},
\end{aligned}
$$

where we know $G$ is invertible because it must be rank $r$ for $B$ to be rank $r$. By a similar argument as above, we can compute $C^{-1}v$ in $O(\alpha n + rn)$ operations as well. $\qquad\square$

**Proposition 1.** *Suppose* $\Sigma_\alpha : \mathbb{R}^{(|T| \times |F| \times |S|) \times (|T| \times |F| \times |S|)}$ *and* $\Sigma_\beta : \mathbb{R}^{(|S| \times |L|) \times (|S| \times |L|)}$ *are covariance matrices arising from the evaluation of kernels (1) and (2) on a grid of points. There exist factorizations* $\Sigma_\alpha = S_\alpha S_\alpha^T$, $\Sigma_\beta = S_\beta S_\beta^T$ *such that multiplying vectors by* $S_\alpha$ *and* $S_\beta$ *(and their inverses) can be performed in* $O\big(|T|\,|F|\,|S|\,(|T| + |F| + |S|)\big)$ *and* $O\big(|S|\,|L|\,(|S| + |L|)\big)$ *arithmetic operations, respectively.*

*Proof.* Since the matrices $\Sigma_\alpha$ and $\Sigma_\beta$ arise from the evaluation of the kernel functions on grids $\{(t,f,s)|t \in T, f \in F, s \in S\}, \{(s,\ell)|s \in S, \ell \in L\}$, we know

$$(\Sigma_\alpha)_{(t_1,f_1,s_1),\,(t_2,f_2,s_2)} = k_\alpha\Big((t_1,f_1,s_1),(t_2,f_2,s_2)\Big)$$
$$= k_T(t_1,t_2)\,k_F(f_1,f_2)\,k_{S_1}(s_1,s_2) + k_{S_2}(s_1,s_2),$$
$$(\Sigma_\beta)_{(s_1,\ell_1),\,(s_2,\ell_2)} = k_\beta\Big((s_1,\ell_1),(s_2,\ell_2)\Big)$$
$$= k_L(\ell_1,\ell_2)k_{S_1}(s_1,s_2).$$

We can thus write

$$\Sigma_\alpha = \Sigma^{(T)} \otimes \Sigma^{(F)} \otimes \Sigma^{(S_1)} + J_{|T| \times |F|} \otimes \Sigma^{(S_2)},$$
$$\Sigma_\beta = \Sigma^{(S_1)} \otimes \Sigma^{(L)},$$

where

$$(\Sigma^{(T)})_{t_1,t_2} = k_T(t_1,t_2), \qquad (\Sigma^{(F)})_{f_1,f_2} = k_F(f_1,f_2), \qquad (\Sigma^{(S_1)})_{s_1,s_2} = k_{S_1}(s_1,s_2),$$

$$(\Sigma^{(S_2)})_{s_1,s_2} = k_{S_2}(s_1,s_2), \qquad\qquad (\Sigma^{(L)})_{\ell_1,\ell_2} = k_L(\ell_1,\ell_2),$$

and $J_{|T| \times |F|} \in \mathbb{R}^{(|T| \times |F|) \times (|T| \times |F|)}$ denotes a matrix of all ones.

Note also that we assume all our kernel functions are positive definite, and so $\Sigma_\alpha$, $\Sigma_\beta$, and all $\Sigma^{(*)}$ factors are positive definite. By properties of Kronecker products, this implies that for any $p \in \mathbb{R}$,

$$\left(\Sigma^{(T)} \otimes \Sigma^{(F)} \otimes \Sigma^{(S_1)}\right)^P = \left(\Sigma^{(T)}\right)^P \otimes \left(\Sigma^{(F)}\right)^P \otimes \left(\Sigma^{(S_1)}\right)^P,$$
$$\Sigma_\beta^p = \left(\Sigma^{(S_1)} \otimes \Sigma^{(L)}\right)^P = \left(\Sigma^{(S_1)}\right)^P \otimes \left(\Sigma^{(L)}\right)^P.$$

Let

$$S_\beta = \Sigma_\beta^{1/2} = \left(\Sigma^{(S_1)}\right)^{1/2} \otimes \left(\Sigma^{(L)}\right)^{1/2}$$

and note that $S_\beta^{-1} = \left(\Sigma^{(S_1)}\right)^{-1/2} \otimes \left(\Sigma^{(L)}\right)^{-1/2}$. We see that $\Sigma_\beta = (S_\beta)^2 = S_\beta S_\beta^T$, and Lemma 2 ensures that we can compute $S_\beta v$ and $S_\beta^{-1} v$ in $O\big(|S|\,|L|\,(|S| + |L|)\big)$ operations for all $v \in \mathbb{R}^{|S| \times |L|}$.

Next let

$$A = \left(\Sigma^{(T)}\right)^{1/2} \otimes \left(\Sigma^{(F)}\right)^{1/2} \otimes \left(\Sigma^{(S_1)}\right)^{1/2}, \qquad\qquad B = J_{|T| \times |F|} \otimes \Sigma^{(S_2)}.$$

As above, Lemma 2 ensures that $Av$ and $A^{-1}v$ can be computed in $O\big(|T|\,|F|\,|S|\,(|T| + |F| + |S|)\big)$ operations for all $v \in \mathbb{R}^{|T| \times |F| \times |S|}$. We further note that $\text{rank}(J_{|T| \times |F|}) = 1$, $\text{rank}(\Sigma^{(S_2)}) = |S|$, and so by properties of the Kronecker product $\text{rank}(B) = |S|$. Since

$$\Sigma_\alpha = AA^T + B,$$

Lemma 3 ensures that there exists a $S_\alpha \in \mathbb{R}^{(|T| \times |F| \times |S|) \times (|T| \times |F| \times |S|)}$ such that $S_\alpha S_\alpha^T = \Sigma_\alpha$ and we can compute $S_\alpha v$ and $S_\alpha^{-1} v$ in $O\big(|T|\,|F|\,|S|\,(|T| + |F| + |S|)\big)$ operations for all $v \in \mathbb{R}^{|T| \times |F| \times |S|}$.
$\qquad\square$