[Reviews · NeurIPS 2018]

Reviewer 1



This paper describes a system for localization and separation of sound sources recorded by distributed microphone arrays. It is similar to MESSL (Mandel et al, 2009) in its building blocks, but builds upon it by employing a Gaussian process prior to encourage smoothness in time, frequency, and location. It also is targeted to the same application as DALAS (Dorfan et al, 2015) of microphones being distributed in pairs around a large space, but adds the smoothness over time and frequency via the Gaussian process prior. It is evaluated on synthetic sound mixtures that are anechoic, but contain a reasonable amount of additive uncorrelated noise. Its performance is quantified using a new metric that compares the masks it estimates at each microphone to the ideal ratio mask at each microphone. Results show that it generally provides a higher level of "power precision" compared to other systems while achieving a similar level of "power recall". Quality: This paper is thorough in its description, derivations, and evaluation. The targeted situation is a little contrived, but the same system seems like it should be able to apply straightforwardly to other situations. The approach is sound and introduces a very nice and flexible framework for incorporating prior assumptions into this model. Evaluation is carried out carefully and shows good results for the proposed system. One relatively minor shortcoming of the paper is that it is not clear that new metrics are necessary for evaluating the performance of this system, and make it slightly harder to contextualize the results. This is especially true with the introduced precision/recall-style metrics which do not provide a single figure of merit with which to compare systems. There are several methods that would be appropriate for comparing synthetic multi-channel inputs and outputs, such as BSS_EVAL or PEASS. Single-channel metrics such as STOI might also be used by finding the best microphone signal to analyze, as was done with the MESSL results in the current paper. There are a improvements to the overall approach that could be addressed in subsequent work: It would also be interesting to use the derived masks to perform beamforming, which would produce a single-channel estimate of each source that could be compared to the original using STOI or subjective evaluations. It would be more realistic to use diffuse noise rather than noise that is uncorrelated across sensors. Diffuse noise is correlated at low frequencies, see for example Habets and Gannot (2007). Habets, E. A., & Gannot, S. (2007). Generating sensor signals in isotropic noise fields. The Journal of the Acoustical Society of America, 122(6), 3464-3470. Clarity: It is well written and generally easy to follow the logic, although many of the details have been compressed to save space, especially in the model description and derivation in section 2. Since the paper provides a sophisticated combination of graphical modeling with microphone array processing, such details could be useful for readers from each of those communities to better understand the other, but it is understandable given the page limit. One specific point that could improve the clarity would be describing the details of the kernels used in equations (1) and (2) adjacent to those equations instead of in the experiments section. The baseline algorithm also performs surprisingly well, for example in figure 2. It would be useful and interesting to describe it in more detail. Minor clarity issues/questions: * On line 138, should C have a subscript associated with it, i.e., should there be different C's for different sources? * On line 141, why is only beta normalized? * On line 149, in most of the paper t_1 appears to be a time index at the frame level, but here it appears to be at the sample level, could this be clarified? Is this smoothing happening between frames? What w(.) function was used in the experiments? Originality: The paper does a good job of describing related work and directly comparing it to the proposed system. In fact, some description of the previous work could perhaps be compressed, since it is discussed in the main introduction, the Prior Work section, and the Experiments section. The proposed approach provides a very nice framework for extending MESSL that can be built upon in the future with additional assumptions and components. Significance: The proposed system provides a interesting improvement of the MESSL approach and paves the way for further improvements. Empirically it seems to show good performance in the setting in which it was evaluated. Response to author feedback: I agree with much of what the authors say in their response, but do not fully agree that SIR and SDR only focus on "precision" of source separation. While there are several incompatible definitions of SIR and SDR in the literature, there are certain existing definitions, for example SNR_ME in [A] that combine both precision and recall by measuring the ratio of the target speech present in the output (precision) to the combination of target speech energy absent from the output (recall) and the noise energy present in the output. BSS_EVAL, while widely used, does in fact focus on precision to the exclusion of recall. [A] Mandel, M. I., Bressler, S., Shinn-Cunningham, B., & Ellis, D. P. (2010). Evaluating source separation algorithms with reverberant speech. IEEE Transactions on audio, speech, and language processing, 18(7), 1872-1883.

Reviewer 2



The paper presents a method for joint source separation and localization based on probabilistic modelling of inter-microphone phase differences and Bayesian inference. The probabilistic method presented for time-frequency mask estimation follows a very similar strategy in several well-known source separation algorithm such as DUET and MESSL. The garbage source idea was also used in MESSL. The novelty of the proposed method is very limited. The results shown in Figure 3 does not show clear advantage as compared with the baseline methods. The subplots in the second line of Figure 2 are not clearly explained. What do you mean by the positions of the sources, what are the horizontal and vertical axes, and how were the plots obtained? There are commonly used performance index for source separation, such as SDR, SIR etc. Why not using these metrics?

Reviewer 3



The authors consider the audio source separation problem in a scenario, where they have access to multiple pairs of omni-directional microphones. They assume that the environment is anechoic, and develop an algorithm that can detect the position of the sources in the environment. A generative model is proposed where a random variable is assigned to each time-frequency-source-space point (the space is partitioned into a grid). The separation is performed by finding source-dependent masks in the STFT domain. I think the anechoic environment assumption is not realistic (outdoors recordings are not affected by reverberation, but they usually contain wind noise), and this significantly reduces the relevance of the proposed formulation/algorithm. In addition, I think the model is rather delicate (assigning a variable for each position in the spatial grid, for instance), which may possibly lead to robustness issues. (The authors also note that the model is fairly sensitive to initialization.) Overall, I think the paper is interesting for its abstract model/algorithm, however, I don't think the considered scenario would be very useful in practice.